# An Artificial Intelligence Approach to Guiding the Management of Heart Failure Patients Using Predictive Models: A Systematic Review

**DOI:** 10.3390/biomedicines10092188

**Published:** 2022-09-05

**Authors:** Mikołaj Błaziak, Szymon Urban, Weronika Wietrzyk, Maksym Jura, Gracjan Iwanek, Bartłomiej Stańczykiewicz, Wiktor Kuliczkowski, Robert Zymliński, Maciej Pondel, Petr Berka, Dariusz Danel, Jan Biegus, Agnieszka Siennicka

**Affiliations:** 1Institute of Heart Diseases, Wroclaw Medical University, 50-556 Wroclaw, Poland; 2Department of Physiology and Pathophysiology, Wroclaw Medical University, 50-368 Wroclaw, Poland; 3Department of Psychiatry, Division of Consultation Psychiatry and Neuroscience, Wroclaw Medical University, 50-367 Wroclaw, Poland; 4Institute of Information Systems in Economics, Wroclaw University of Economics and Business, 53-345 Wroclaw, Poland; 5 Department of Information and Knowledge Engineering, Prague University of Economics and Business, W. Churchill Sq. 1938/4, 130 67 Prague, Czech Republic; 6Department of Anthropology, Ludwik Hirszfeld Institute of Immunology and Experimental Therapy, Polish Academy of Sciences, 53-114 Wroclaw, Poland

**Keywords:** artificial intelligence, machine learning, deep learning, heart failure, predictive model, systematic review

## Abstract

Heart failure (HF) is one of the leading causes of mortality and hospitalization worldwide. The accurate prediction of mortality and readmission risk provides crucial information for guiding decision making. Unfortunately, traditional predictive models reached modest accuracy in HF populations. We therefore aimed to present predictive models based on machine learning (ML) techniques in HF patients that were externally validated. We searched four databases and the reference lists of the included papers to identify studies in which HF patient data were used to create a predictive model. Literature screening was conducted in Academic Search Ultimate, ERIC, Health Source Nursing/Academic Edition and MEDLINE. The protocol of the current systematic review was registered in the PROSPERO database with the registration number CRD42022344855. We considered all types of outcomes: mortality, rehospitalization, response to treatment and medication adherence. The area under the receiver operating characteristic curve (AUC) was used as the comparator parameter. The literature search yielded 1649 studies, of which 9 were included in the final analysis. The AUCs for the machine learning models ranged from 0.6494 to 0.913 in independent datasets, whereas the AUCs for statistical predictive scores ranged from 0.622 to 0.806. Our study showed an increasing number of ML predictive models concerning HF populations, although external validation remains infrequent. However, our findings revealed that ML approaches can outperform conventional risk scores and may play important role in HF management.

## 1. Introduction

Heart failure (HF) remains a major clinical and public health problem. HF is associated with substantial morbidity and mortality but also with poor quality of life. HF affects more than 64 million people worldwide [1]; its prevalence is estimated at 1–3% in the general population and is expected to rise due to the ageing of the population and improved survival of treated patients. In consequence, HF represents 1–2% of all hospitalizations and is still a leading cause of admissions in Europe and the United States [2]. Given that, risk stratification and prognosis prediction in HF populations is significant. Risk assessments play a crucial role in identifying high-risk cases and in guiding clinical decisions. A tailored, patient-level approach improves survival and quality of life and could reduce the rate of readmissions and the burden on the health care system. However, precisely predicting outcomes in heart failure patients remains difficult. There is a great need to develop and validate data-driven predictive models supporting this purpose.

Recently, artificial intelligence (AI) methods are successfully implemented in several medical fields e.g., in radiological images analysis or in prediction of suicide attempts [3,4,5,6,7,8,9,10]. The same applies to heart failure population. The clustering technology enables classification of HF patients with regard to their clinical characteristic [11,12,13,14]. Machine learning techniques provides tools to discriminate HF patients from subjects with no HF, where most of current models use heart rate variability to detect heart failure [15,16,17,18,19]. One of the widely known problem in clinical practice is accurate selection of candidates for cardiac resynchronization therapy (CRT). The high percentages of nonresponders for CRT remains an important problem. ML methods showed the possibility of improving decision making in CRT [20,21,22,23]. ML approaches can be also used to predict untypical outcomes such as treatment adherence [24] and left ventricular filling pressure among HF patients [25] and can reveal relationships between HF symptom profiles and depressive symptoms [26]. Finally, AI algorithms can predict crucial outcomes in HF management such as mortality and readmission rates. Currently, clinicians have at their disposal several predictive models focusing on heart failure such as Get with the Guide- lines-Heart Failure (GWTG-HF) score and Meta-Analysis Global Group in Chronic Heart Failure (MAGGIC) score [27,28] (Table 1). These models used conventional statistical approaches, mostly multivariate analysis based on logistic regression. These scores have several limitations such as inability to capture multidimensional correlations between variables in medical records. Moreover, their usefulness can be hampered in specific situations; they were developed on selected cohorts and therefore may achieve only modest accuracy. Furthermore, the clinical implementation of these tools is limited due to the requirement of medical staff involvement in each patient’s risk estimation.

Conversely, AI algorithms seem to have certain advantages in these fields. Machine learning (ML) algorithms are able to capture nonlinear, unstructured interactions between the data, including clinical features, and their associations with a patient’s prognosis [29]. Thus, this type of approach can achieve superior accuracy compared with the linear models. ML-based predictive models provide an opportunity for more individualised, patient-level management. Of note, a significant increase in the number of studies which included AI-based predictive models in medicine has been reported recently [30]. The role of this novel approach will be increasing in clinical practice in the near future. Hence, this study aims to screen and analyse predictive models based on AI algorithms among HF patients. We took into consideration all types of outcomes: mortality, rehospitalization, response to treatment and medication adherence among patients with already detected HF. To the best of our knowledge, this is the first systematic review concerning ML predictive models with external validation.

## 2. Methods

### 2.1. Search Strategy

A systematic literature review was guided by the Preferred Reporting Items for Systematic Reviews and Meta-Analyses (PRISMA) statement [31]. Two independent reviewers (M.B. and S.U.) engaged in online searches, and all disputable issues were solved through discussion with the third reviewer (M.J.). Literature searches were conducted in Academic Search Ultimate, ERIC, Health Source Nursing/Academic Edition and MEDLINE until 31 March 2022. In addition, the references of eligible reports and papers which cited the included studies were screened. The protocol of the current systematic review was registered in the PROSPERO database with the registration number CRD42022344855. In order to find the relevant studies, we used the following combination of keywords: Mortal* OR Death OR Readmi* OR Rehosp* OR Hospital* OR Risk* OR Predict* OR Prognos* OR Admission* OR Outcome*) AND (Machine Learning OR Deep Learning OR artificial intelligence OR artificial neural network* OR Unsupervised OR supervised) AND (Heart Failure OR Heart Failure Systolic OR Heart Failure Diasystolic OR Acute Heart Failure OR AHF OR Chronic Heart Failure OR CHF OR Heart Failure with preserved ejection fraction OR HFpEF OR Heart Failure with mid-range ejection fraction OR HFmrEF OR Heart Failure with reduced ejection fraction OR HFrEF).

### 2.2. Eligibility Criteria

All records that incorporated machine learning and/or deep learning models among patients with heart failure (risk prediction of readmission after index hospitalization, risk of mortality, prediction of response to treatment and medication adherence) were considered important and included in our systematic review. Other inclusion criteria were: external validation of the model and confirmed heart failure. Analysed records were excluded from the systematic review if they met the following conditions: internal validation only, implemented logistic regression alone, models predicted first incidence of HF, congenital HF was a studied population, unconfirmed heart failure, reviews, commentaries, editorials, not available in English language, did not have full text available and animal model studies.

### 2.3. Data Extraction and Quality Assessment

The following data were extracted from eligible publications: (1) year of publication, (2) data source, (3) size of training cohort, (4) size of validation cohort, (5) setting, (6) outcomes, (7) predictor variables, (8) algorithm used, (9) AUC, (10) comparator score in the particular dataset and (11) management of missing values. Quality assessment was performed using a checklist for quantitative studies [32]. This checklist contains 14 questions, and evaluation was performed by two reviewers (M.B. and M.J.). The total score of this scale range between 0 and 28. In the first step, the mean of the two assessments was calculated. In the next step, quality scores were expressed as the percentage of maximum quality score which can be assigned to a particular study. A higher percentage indicates a better-quality study (Table 2).

## 3. Results

### 3.1. The Review Process

The initial search identified 1297 publication records through database searching and 352 through other sources (Figure 1). After screening titles and abstracts, only 9 studies met the inclusion criteria and did not meet exclusion criteria. The excluded publications were: 307 duplicates, 578 based on a different model, 159 different studied populations, 73 reviews, 13 systematic reviews, 29 editorials, 4 animal studies, 8 posters, 4 lectures and 7 that were not available in English. What we mean by different model is studies using conventional, statistical methods or an AI analysis other than prediction. In the next step, the full texts of 115 studies were assessed concerning type of validation. External validation was missing in 106 studies, which were excluded. Additionally, we analysed the references of eligible publication records and studies that cited the included papers. None of the 352 screened records met the inclusion criteria. Finally, nine predictive models were included in the systematic review (Table 2).

### 3.2. Comparison of the Predictive Value

We aimed to thoroughly describe and compare the predictive value of the different models, and thus, we decided to use the area under the receiver operating characteristic curve (AUC) as a standard parameter. The AUC is commonly used to compare the performance of models with each other and with conventional prediction scores. The AUC represents the sensitivity against 1-specificity (Figure 2). The AUC of a classifier means the probability that the classifier model will rank a randomly chosen positive instance higher than a randomly chosen negative instance. The AUC is always between 0 and 1.0, where an AUC of 0.5 means no better accuracy than chance and an AUC of 1.0 means perfect accuracy [42]. To compare the certain AUC scores, we can refer to the following rule: AUC = 0.5 means no discrimination, AUC = 0.5–0.7 means poor discrimination, AUC = 0.7–0.8 means acceptable discrimination, AUC = 0.8–0.9 means excellent discrimination, AUC > 0.9 means outstanding discrimination [43].

### 3.3. Relevant Studies

The first study conducted by Luo et al. aimed to create a risk stratification tool assessing the all-cause in-hospital mortality in intensive care unit (ICU) patients with HF [33]. The model was developed based on records from the Medical Information Mart for Intensive Care MIMIC-III database and externally validated with the use of eICU database. The study included patients admitted to the ICU due to their first manifestation of HF. The demographic data and comorbidities were collected at admission, and then vital signs and laboratory records were collected hourly during the first 24 h after admission. In this way, physicians took the minimum, maximum, mean and range of the values over a period. After records extraction, the variables with missing data of more than 40% and patients with more than 20% missing parameters were excluded to avoid bias. The XGBoost algorithm was used to develop the machine learning model. The derivation data (5676 patients) were randomly divided into a training cohort (90%), and then the rest of the cohort (10%) was used to validate the performance. Finally, 24 features were selected as the most important from the predictive model as follows: mean anion gap, mean Glasgow Coma Scale, urine output, mean blood urea nitrogen (BUN), maximum Pappenheimer O2 (pO2), age, mean plasma calcium, minimum plasma glucose, mean plasma magnesium, mean respiratory rate (RR), mean arterial base excess, mean creatinine, body mass index (BMI), mean temperature, maximum temperature, maximum platelet, minimum prothrombin time (PT), mean systolic blood pressure (SBP), mean partial thromboplastin time (PTT), mean oxyhaemoglobin saturation (spO2), mean PT, mean diastolic blood pressure (DBP) and minimum PTT. In internal validation (10% of the derivation data), the AUC reached 0.831. In the next stage, records from the eICU database were used to conduct external validation; the AUC was 0.809. In effect, the current classifier had only a slight deterioration in performance in the external cohort. Anion gap, blood coagulation status and volume of urine output were found to be the three most important predictors in this model.

Kwon et al. described a deep-learning-based artificial intelligence algorithm for predicting mortality of patients with acute heart failure (DAHF) [35]. The endpoints were in-hospital, 12-month and 36-month mortality. The patients’ electronic health records (demographic information, treatment therapy, laboratory results, electrocardiography (ECG) and echocardiography results, final diagnosis and clinical outcomes during their hospital stay) were collected from two hospitals during their admission and used to train the algorithm (2165 patients). The deep neural network was used to develop a prediction model. Further, separate data from the independent Korea Acute Heart Failure register (KorAHF) with 4759 patients were implemented as an external validation cohort. The age, sex, body mass index, SBP, DBP, heart rate (HR), present atrial fibrillation, QRS duration, corrected QT interval, left atrial dimension, left ventricular dimension end-diastole, left ventricular dimension end-systole, ejection fraction (EF), white blood cell (WBC), haemoglobin, platelet, albumin, sodium, potassium, blood urea nitrogen, creatinine and glucose were assessed as the predictor variables. These data were obtained during each ECG assessment, and in consequence, complete datasets for each patient were created. The AUCs for DAHF were 0.880 for predicting in-hospital mortality and 0.782 and 0.813 for predicting 12- and 36-month mortality, respectively, in the KorAHF register. The well-established scores revealed poorer AUCs: 0.782 (GWTG-HF) for in-hospital mortality, 0.718 for 12-month mortality and 0.789 for 36-month mortality (MAGGIC).

Another study by Kwon et al. aimed to develop a machine learning predictive model for mortality among heart disease patients based only on the results of echocardiography [38]. The data from the first hospital (20,651 patients) were used as the derivation data and for internal validation (by splitting), and data from the second hospital (1560 patients) were used for external validation. The derivation data consists of patients with atrial fibrillation/flutter (AF/AFL), HF and coronary artery diseases (CAD). The internal validation group contained 3840 subjects from the first hospital. The external validation group consisted of 604 subjects with CAD and 760 subjects with HF. Patients with missing values were excluded. The primary outcome was in-hospital mortality. Only echocardiography features were used as predictor variables: 11 continuous (age, weight, height, HR, left ventricular diastolic diameter (LVDD), left ventricular systolic dysfunction (LVSD), septum thickness, posterior wall thickness (PWT), aorta dimension, left atrium dimension and EF) and 54 categorical (rhythm, mitral valve description (10 features), aortic valve description (9 features), mitral valve description (9 features), left ventricle regional description (12 features), left ventricle functional description (2 features), pericardium description (2 features), inferior vena cava (2 features) and right ventricle description (2 features)). The deep neural network (DNN) was used to develop a prediction model. In internal validation, the model yielded AUC = 0.912 for predicting in-hospital mortality for heart diseases. In external validation, the model achieved AUC = 0.898 for heart diseases and 0.958 for CAD. In the HF group, the created model produced AUC = 0.913 in external validation in comparison with the MAGGIC score (0.806) and GWTG-HF (0.783). The ML model outperformed the existing predictive models regardless of the underlying disease.

The machine learning assessment of risk and early mortality in HF (MARKER-HF) risk scale was developed based on a cohort of 5822 patients from out- and inpatient care. They were identified from medical history by their first episode of HF [34]. The boosted decision tree algorithm was used to build the model. During the training process, eight variables were identified as the predictor features: DBP, creatinine, BUN, haemoglobin (Hb), WBC count, platelets, albumin and red blood cell distribution width (RDW). The external validation was performed with the use of two, independent registers containing 1512 and 888 subjects. The model was designed to distinguish patients with high and low risk of death. The patients who died before 90 days since the index hospitalization were considered the high-risk group, and patients with a last-known follow-up 800 or more days after the index hospitalization were classified as the low-risk group. In internal validation, AUC = 0.88, while external validation in two independent cohorts gave AUCs of 0.84 (888 patients) and 0.81 (1516 patients). Moreover, the authors compared the performance of the model with the levels of N-terminal pro-B-type natriuretic peptide (NT-proBNP), which were available in the derivation database. NT-proBNP is a well-established biomarker associated with mortality amongst HF patients [45,46]. A higher MARKER-HF score is associated with high NT-proBNP, but in the comparison of predictive power, MARKER-HF reached superior AUC to that of natriuretic peptide (0.88 vs. 0.69). In comparison with the GWTG-HF (AUC = 0.74), the created model presented superior discriminatory power in all three populations.

Another study conducted by Chirinos et al. concerns associations between plasma biomarkers of patients with heart failure with preserved ejection fraction and the composite endpoint of all-cause death or heart failure-related hospital admission [37]. The authors selected 379 patients from the Treatment of Preserved Cardiac Function Heart Failure with an Aldosterone Antagonist Trial (TOPCAT) database for creating their predictive model, and they validated it externally (156 subjects) with the use of data from the Penn Heart Failure Study (PHFS). Only patients with fully available variables from TOPCAT were included in the study. The tree-based pipeline optimizer platform was utilized to build a predictive model, and the following biomarkers were found to be relevant for predicting death or HF-related rehospitalization: 2 biomarkers related to mineral metabolism/calcification (fibroblast growth factor 23 (FGF-23) and osteoprotegerin (OPG)), 3 inflammatory biomarkers (tumour necrosis factor-alpha (TNF-alpha), soluble tumour necrosis factor receptor I (sTNFRI) and interleukin 6 (IL-6)), YKL-40 (related to liver injury and inflammation), 2 biomarkers related to intermediary metabolism and adipocyte biology (fatty acid-binding protein 4 (FABP-4) and growth differentiation factor 15 (GDF-15)), angiopoietin-2 (related to angiogenesis), matrix metallopeptidase 7 (MMP-7, related to extracellular matrix turnover), ST-2 cardiac biomarker and NT-proBNP. In this project, test performance was assessed with the C-index (concordance index), which is analogous to the receiver operator characteristic curve. In the internal validation, the C-index was 0.743, whereas in the external validation was 0.717. Moreover, the authors combined their ML model with the MAGGIC score. As a result, the C-index for this combination was 0.73 in internal and external validations. There was a slight deterioration in the C-index in the external validation of the machine learning (ML) model alone, but the model containing ML and the MAGGIC score revealed the same C-index in both cohorts. This shows the very similar predictive power between the ML model alone and the model containing ML and the MAGGIC score. The FGF-23, YKL-40 and sTNFRI were found to be the three biomarkers most associated with the endpoint.

Jing et al. created a ML model for predicting 1-year all-cause mortality among HF patients [36]. The data from 26,971 subjects (with 276,819 clinical episodes) were used to train the model, and data from 548 patients/episodes were used to perform external validation. All clinical visits since 6 months before HF detection date including outpatient visits, hospitalizations, emergency department admissions, laboratory tests and cardiac diagnostic measurements were identified and grouped into episodes. All clinical visits since 6 months before the HF diagnosis date, which includes outpatient visits, hospitalizations, emergency department admissions, laboratory tests and cardiac diagnostic measurements were identified, grouped into episodes and used as independent sets. The predictive model was based on the XGBoost algorithm. The following features were incorporated from the electronic health records: 26 clinical variables (age, sex, height, weight, smoking status, HR, SBP, DBP, use of loop diuretics, antihypertensive and antidiabetic medications and laboratory test values (haemoglobin, estimated glomerular filtration rate, creatine kinase-muscle/brain, lymphocytes, high-density lipoprotein, low-density lipoprotein, uric acid, sodium, potassium, NT-proBNP, troponin T (cTnT), haemoglobin A1c (HbA1c), troponin I (cTnI), creatinine, and total cholesterol), 90 cardiovascular diagnostic codes (International Classification of Diseases-10th—ICD10), 41 ECG assessments and patterns, 44 echocardiographic measurements and 8 evidence-based “care gaps”: flu vaccine, blood pressure of < 130/80 mm Hg, HbA1c of < 8%, cardiac resynchronization therapy, and active medications (active angiotensin-converting enzyme inhibitor/angiotensin II receptor blocker/angiotensin receptor-neprilysin inhibitor, aldosterone receptor antagonist, hydralazine, and evidence-based beta-blocker). The authors selected these care gaps as the important evidence-based interventions in HF treatment for which associations with risk reduction were assessed. To measure the predicted effect of closing care gaps on reducing rate of endpoint, the authors closed artificial care gaps according to the following formula: for binary gap variables, changing the value from 1 (open/untreated) to 0 (closed/treated); for continuous variables, changing the value to goal. After that, the influence of this simulation on the risk score was calculated. Only complete sets of variables were included. The model achieved discriminatory power in assessing mortality risk with an AUC of 0.77 in cross-validation and 0.78 in the external validation. This showed that the model tended to slightly overestimate the risk of mortality. Moreover, the simulation of closing the 8 care gaps resulted in a 1.7% reduction of mortality.

Mahajan et al. developed two predictive models using different ML methods. The first of them combined structured and unstructured data, and the second one used ensemble ML methods for predicting the risk of readmissions for HF. All dependent variables were available, and there were up to 5% missing values of independent variables; thus, in order to maintain consistency, the authors used multiple imputation by chained equations resampled over five imputed datasets for the missing values assuming missingness at random. Both models aimed to predict 30-day readmissions, and both studies used the same structured data predictors: sodium, potassium, BUN, creatinine, Hb, haematocrit (Ht), glucose, albumin, B-natriuretic peptide, SBP, DBP, pulse, RR, demographics (age, sex, race, marital status, insurance type, residential area), pre-index admission factors (appointments in past year, no show to appointment in past year, emergency department visits in past year, prior diagnoses, admissions in previous year, telemetry monitor during index admission, index admission via emergency length of stay concurrent procedures), comorbidities (alcohol abuse, cardiac arrhythmia, CAD, cancer, cardiomyopathy, cerebrovascular accident, depression, diabetes mellitus, drug abuse, functional disability, liver disease, lung disease, protein caloric malnutrition, psychiatric disorder, rheumatic disease group, renal disease group, vascular disease group, aortic valve disorder), and concurrent procedure (cancer related, cardiac devices, cardiac surgery, coronary angioplasty, history of mechanical ventilation devices). In the first instance, the authors used the parametric statistical method and statistical natural language processing (NLP) to create three models: one using structured, one using unstructured data and one that combined these two approaches [40]. The structured dataset contained 1619 patients, where 1279 patients enrolled from 2011 to 2014 were the derivation cohort, and 340 patients from 2015 were the external validation subgroup. Then, 136,963 clinical notes were extracted (as unstructured records) such as history and physician notes at admission, progress notes, social workers’ notes and discharge summaries. Of these, 102,055 notes were for the derivation cohort, and 34,908 were for the validation cohort. The combined dataset had over 4900 predictors, although the authors showed only 10 with relative importance: creatinine, BUN, haematocrit and other not listed administrative predictors. The AUC for the structured model was 0.6494, for the unstructured model, 0.5219 and for the combined model, 0.6447. As a result, the performance of the structured and combined models was very similar, but the unstructured data model showed very poor discrimination. In the second instance, the authors selected 27,714 admissions (from 2011 to 2014) that represented a derivation cohort, and 8531 admissions from 2015 were used as the external validation subgroup [39]. The authors used 10 different base learning models and two ensemble schemes to combine base learner outputs (Super learner and Subsemble scheme). Further, the AUCs for each base learner and ensemble schemes were calculated. The best single base learner achieved AUC = 0.6993 (Extratrees); for Super Learner, it was 0.6987 and for Subsemble, 0.6914. This showed that ensemble techniques can ensure performance at least as good as the best-performing single-base algorithm.

The protocol of the study conducted by Kakarmath et al. presents a promising design for investigations [41]. This project aimed to build a ML model predicting 30-day readmissions in HF patients. The study concerns all types of heart failure: left; systolic, diastolic, combined; acute, chronic, acute on chronic and unspecified with the expected population of 1228 index admissions.

## 4. Discussion

Our systematic review revealed several factors with significant impacts on the utility of AI-based tools in heart failure patient management. First, this study showed an increasing number of studies concerning artificial intelligence methods incorporated in heart failure population (Figure 3). Particularly during the last four years, we can observe a significant increase of interest in this field.

Second, this analysis has shown that tens of predictive models are being generated, but only a small part of them were externally tested. External validation means the assessment of predictive performance (discrimination and calibration) with the use of an independent, individual dataset different from the one used to generate the model [47]. The external validation can be performed in different cohorts including race, geographical region, period, social-economic settings or type of care (outpatient/inpatient) [48]. This approach determines objective discriminating ability in different settings from those of the derivation data, thus revealing the utility of the model in the real world. The evaluation of the model based only on the derivation data can lead to misleading performance, for example, mortality probability models (MPM II) predicting mortality among intensive care unit patients achieved AUC = 0.836 [49], whereas the external validation performed 16 years later revealed only a modest AUC of 0.66, demonstrating very low discriminating ability [50]. In the case of HF, it is worth mentioning that this disease is characterized by different causes, comorbidities and demographic profiles around the world [51,52]. Given that, the lack of test performance in different circumstances is the first barrier to applying prediction calculators in clinical practice.

Third, our analysis revealed that machine learning predictive models can accurately predict different types of outcomes among HF populations. This is particularly important when we compare the performance of AI-based models with conventional statistical predictive models [53]. Most of the selected studies in our review showed satisfactory accuracies in external validation. The AUCs range from 0.6494 to 0.913 in independent datasets, whereas AUCs for statistical predictive scores range from 0.622 to 0.806 (Table 3). What is more, the majority of included studies used either tree-based or neural network-based methods to generate their models. The early assessment of a patient’s clinical risk is crucial for tailored treatment and improving the patient’s prognosis. Furthermore, we revealed that these models can use various types of data to predict the outcomes. This indicates that all types of clinical data such as demographic, laboratory, clinical examination, medication and echocardiographic and electrocardiographic metrics contain predictive information and are suitable for building an effective model. Our study showed the potential of the incorporation of features into predictive models that were not previously considered predictor parameters. It is especially important to use commonly available features in order to provide real-time evaluation and to ensure inclusiveness by avoiding subjects with missing data.

However, ML-based approaches are not free from limitations. First, predictive models are associated with the overfitting problem. Overfitting can lead to the over-training of the training data and in consequence limit discriminatory ability in other populations. One of the solutions is to evaluate the model in an independent cohort. That is why we established external validation as an inclusion criterium. Second, the interpretability or explainability of the created models has become a significant topic in the area of machine learning. The end users are interested not only in the quality of the models but also in understanding the classification processes. Some models are easy to understand by their nature (typical examples are decision trees), but some models, typically neural networks, work as black-box models. To provide some insights into a particular model’s decisions, several approaches have been proposed. Local interpretable model-agnostic explanations (LIME) is an interpretability surrogate model which can be used on any black-box model to provide local interpretability for the prediction or classification of a single instance [54]. SHapley Additive exPlanations (SHAP) is a game theory-based method of interpreting any machine learning model’s output [55]. It uses the traditional Shapley values from game theory and their related extensions to correlate optimal credit allocation with local explanations. As the explainability issue was not discussed in the reviewed papers, we cannot assess them from this point of view.

Third, the described models used only variables that were collected in datasets used as derivation cohorts. Three of all the included studies used a well-established prognostic marker such as NT-proBNP as a predictive feature, but only one model used troponin to predict outcomes [56,57]. There exist other, not included parameters that can be also important prognostic factors for cardiovascular diseases and can potentially improve model performance [58,59].

There is space for applying complex ML techniques in HF management. In the first step, during hospitalization, low-cost and non-invasive ML calculators can be used for patient risk assessments. In the second step, ML models can be used in outpatient care as everyday support tools for patients. The absence of sufficient education, inadequate self-care and lack of medication adherence are well-studied reasons why patients do not reach clinical goals, leading to readmissions, and all these issues can be addressed using ML-based solutions [60]. Personal health plans using sensing devices and artificial intelligence methods can support patients during routine care [61]. The combined approach using ML tools during hospitalization and in outpatient care may result in clinical benefits.

The perspective of AI-based techniques holds great promise for cardiovascular research and clinical practice. Some fields seem especially attractive. Well-trained models can perform real-time, automatized patient risk stratification undertaken at the beginning of the diagnostic path. We can imagine a case when the admitted patient is classified into one of the pre-established clusters. Affiliation with one of the clusters initially suggests diagnostics, treatment and risk groups. Basing patients’ management on models trained on large cohorts could help physicians not to overlook some important issues, especially in conditions of overwork and lack of personnel. Further, AI tools can provide more information about the patient from the same amount of data. Sengupta et al. created a model which can provide the same amount of information about aortic stenosis severity as if both echocardiography and cardiac CT/MR were performed using only echocardiography and the model. Such solutions can be widely implemented in less-developed areas where sophisticated diagnostic methods are not available [62]. Another field that represents great potential is predicting pathology from seemingly physiological findings; for instance, there are successful attempts to identify patients with a history of atrial fibrillation from the ECG performed during the sinus rhythm [63]. The earlier detection of atrial fibrillation can lead to vast reductions in vascular changes in the brain and thereby improve patients’ quality of life.

## 5. Limitations

Our systematic literature review has several limitations. Our selected inclusion and exclusion criteria resulted in a low number of analysed studies. All of the eligible records were markedly heterogeneous in terms of outcomes but did use ML methods and predictor variables. As a consequence, we did not perform meta-analysis, which could have been challenging and unrepresentative. However, meta-analysis would have enabled better insight into the models’ performance, and this issue should be addressed in the next research investigating this field. Finally, our review focuses mainly on the medical issues of predictive models in HF. We aimed to show state-of-art achievements in this area, current solutions and level of advancement. We did not consider technical nuances of modelling processes such as hyperparameter optimization or the tuning of algorithms. There is an increasing need for a pragmatic framework for evaluating clinical utility and validity in ML studies [64]. The development of guidelines for ML implementations in medical research is a clear field for further investigations.

## 6. Conclusions

The implementation of artificial intelligence methods in heart failure management is still in the very early stage. There is a great necessity to evaluate new predictive algorithms, train models in different populations and try to combine various types of predictor variables. Our study showed that artificial intelligence techniques may play an imminent role in heart failure management. Data-driven predictive models showed promise in handling large volumes of medical data. Machine learning techniques may also enable patient-level management, allowing for possibly reducing adverse outcomes in this population.

## Figures and Tables

**Figure 1 biomedicines-10-02188-f001:**
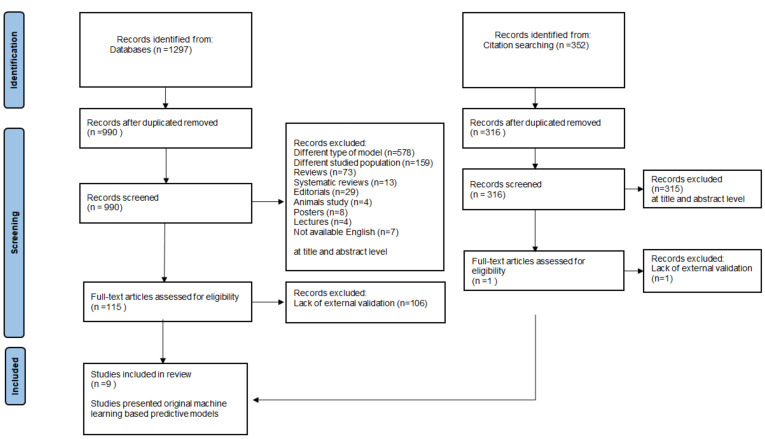
Flow chart of the systematic literature research.

**Figure 2 biomedicines-10-02188-f002:**
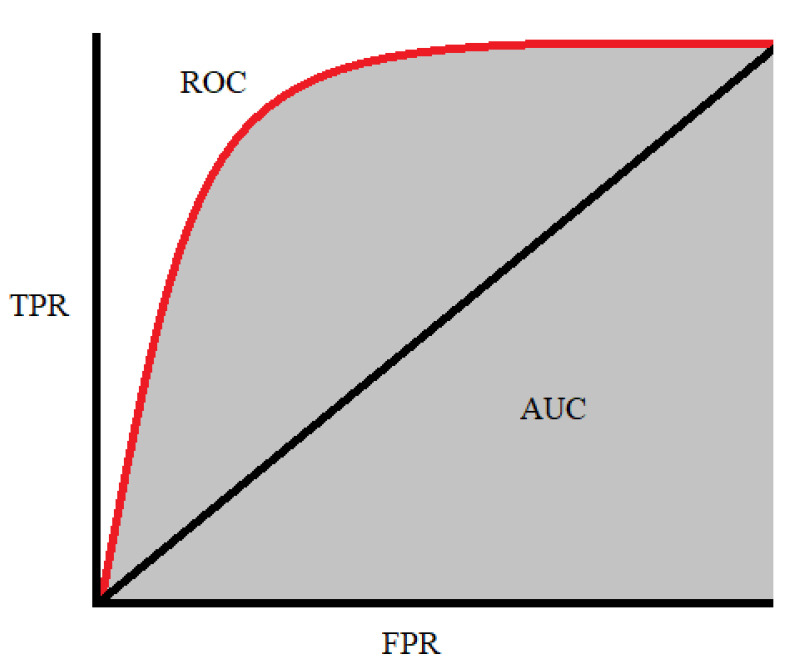
The receiver operating characteristic curve is plotted with the true positive rate (TPR) against the false positive rate (FPR), where FPR is on the *x*-axis and TPR is on the *y*-axis [44].

**Figure 3 biomedicines-10-02188-f003:**
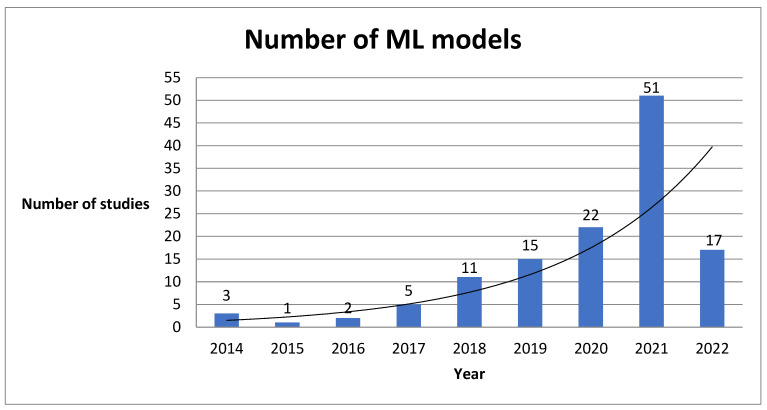
Number of ML predictive models created during last 9 years.

**Table 1 biomedicines-10-02188-t001:** Predictor variables used in MAGGIC and GWTG-HF scores. COPD—Chronic Obstructive Pulmonary Disease, NYHA—New York Heart Association, ACEI—angiotensin-converting enzyme inhibitor, ARB—angiotensin receptor blockers, BMI—body mass index, BUN—blood urea nitrogen.

No.	Variables	MAGGIC	GWTG-HF
1.	Age	X	X
2.	COPD history	X	X
3.	Systolic blood pressure	X	X
4.	Gender	X	
5.	Diabetes	X	
6.	Heart failure diagnosed within the last 18 months	X	
7.	Current smoker	X	
8.	NYHA class	X	
9.	Receives beta blockers	X	
10.	Receives ACEI/ARB	X	
11.	BMI	X	
12.	Creatinine	X	
13.	Ejection fraction	X	
14.	BUN		X
15.	Sodium		X
16.	Black race		X
17.	Heart rate		X

**Table 2 biomedicines-10-02188-t002:** Characteristic of included ML predictive models. MIMIC—Medical Information Mart for Intensive Care, UCSD—University of California, San Diego, KorAHF—Korea Acute Heart Failure, EHR—electronic health record, TOPCAT—Treatment of Preserved Cardiac Function Heart Failure with an Aldosterone Antagonist Trial.

No	Author	Year	Data Source	Population	Setting	Validation Size	Number of Variables	Outcome	Quality
1.	C. Luo et al. [33]	2022	MIMIC-III database	5676	inpatient	1349	24	in-hospital mortality in intensive care unit	87%
2.	E. Adler et al. [34]	2020	UCSD database	5822	inpatient and outpatient	1512 + 888	8	general mortality	80%
3.	J. Kwon et al. [35]	2019	KorAHF	2165	inpatient	4759	23	in-hospital, 12-month and 36-month mortality	87%
4.	L. Jing et al. [36]	2020	Geisinger EHR	26,971	inpatient and outpatient	548	209	1-year all-cause mortality	73%
5.	J. Chirinos [37]	2020	TOPCAT	379	inpatient and outpatient	156	12	composite endpoints of death or heart failure–related hospitalization	87%
6.	J. Kwon et al. [38]	2019	register	20,651	inpatient	1560	65	in-hospital mortality	87%
7.	S. Mahajan [39]	2019	register	27,714	inpatient	8531	Not reported	30-day readmission	66%
8.	S. Mahajan [40]	2019	register	1279	inpatient	340	Not reported	30-day readmission	66%
9.	S. Kakarmath et al. [41]	The protocol for the study							

**Table 3 biomedicines-10-02188-t003:** Performance metrics for machine learning algorithms and conventional risk scores. IV—internal validation, EV—external validation, ML—machine learning, NLP—natural language processing, HF—heart failure, HFrEF—heart failure with reduced ejection fraction, HFmrEF—heart failure with mid-range ejection fraction, HFpEF—heart failure with preserved ejection fraction HD—heart disease, CAD—coronary artery disease.

No.	Author	Algorithm	AUC for ML in IV	AUC for MAGGIC in IV	AUC for GWTG-HF in IV	AUC for ML in EV	AUC for MAGGIC in EV	AUC for GWTG-HF in EV
1.	C. Luo et al. [33]	XGBoost	0.831	-	0.667	0.809	-	-
2.	E. Adler et al. [34]	boosted decision tree	0.88	-	0.74	0.81–0.84	-	0.758
3.	J. Kwon et al. [35]	deep neural network	-	-	-	0.88—in-hospital mortality 0.782—12-month mortality 0.813—36-month mortality	0.718—12-month mortality 0.729—36-month mortality	0.728—in-hospital mortality
4.	L. Jing et al. [36]	XGBoost	0.77	-	-	0.78	-	-
5.	J. Chirinos [37]	created by the tree-based pipeline optimizer platform	0.743 (C-index)	0.621 (C-Index)	-	0.717 (C-index)	0.622 (C-index)	-
6.	J. Kwon et al. [38]	deep neural network	0.912—(HD)	-	-	0.913 (HF) 0.898 (HD) 0.958 (CAD)	0.806 (HF)	0.783 (HF)
7.	S. Mahajan [39]	ensemble ML	-	-	-	0.6987	-	-
8.	S. Mahajan [40]	created by NLP process	-	-	-	0.6494	-	-
9.	S. Kakarmath et al. [41]	The protocol for the study						

## Data Availability

The data presented in this study are available within the article. Further data are available on request from the corresponding author.

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
