# Peer review of "An Artificial Intelligence Approach to Guiding the Management of Heart Failure Patients Using Predictive Models: A Systematic Review"

_biomedicines, 2022, doi:10.3390/biomedicines10092188_

Round 1

Reviewer 1 Report

This manuscript proposed a review for using AI in building the management of heart failure (HF) patients. It is interesting to have related literature discussing the current process, techniques, and limitations of using AI for heart failure prediction and management.  The major concerns are listed below:

1. The review is very interesting, however, it lacks analysis either from width or depth in both clinical and technique perspectives.  I recommend getting into either widther or deeper from the introduction to the results.  For example, could you get more particular clinical issues related to heart failures or HF management?  I am really interested in more introduction and discussion in the clinical parts. I guess that some readers also like more about the current AI techniques in HF. 

2. Can you discuss more the limitations of the current study?  It would be much better to tell what we have not or could not expect from this work.

3.  It is really interesting to show what your view about ML or AI in HF could be in the future. 

4. More references may need and look at line 58 "e.g.:"  is this correct ?

Author Response

We would like to thank the Editors and the Reviewers for their valuable comments. We have amended our manuscript accordingly. The detailed responses to all issues raised by the Reviewers are provided below.

Reviewers' comments:

This manuscript proposed a review for using AI in building the management of heart failure (HF) patients. It is interesting to have related literature discussing the current process, techniques, and limitations of using AI for heart failure prediction and management.  The major concerns are listed below:

  1. The review is very interesting, however, it lacks analysis either from width or depth in both clinical and technique perspectives.  I recommend getting into either widther or deeper from the introduction to the results.  For example, could you get more particular clinical issues related to heart failures or HF management?  I am really interested in more introduction and discussion in the clinical parts. I guess that some readers also like more about the current AI techniques in HF. 

We appreciate this important comment, that AI implementation in HF is not limited only to predictive models. We developed paragraph about other AI utility in HF management. Additionaly, we added section in the discussion about technical issue regarding interpretability or explainability of the created models. This topic remains important problem within the context of reliable assessment of ML studies.

Our adjustment:

# Recently, artificial intellgence (AI) methods are succesfully implemented in several medical fields e.g in radiological images analysis or in prediction of suicide attempts [5-6] [27-32]. The same applies to heart failure population. The clustering technology enables classification of HF patients with regard to their clinical characteristic [44-47].Machine learning techniques provides tools to discriminate HF patients from subjects with no HF, where most of current models use heart rate variability to detect heart failure [48-52]. One of the widely known problem in clinical practice is accurate selection of candidates for cardiac resynchornizatory therapy (CRT). High percetange of nonresponders for CRT remains important problem. ML methods showed possibility to improve decision-making in CRT [53-56]. ML approach can be also used to predict untypical outcomes such as prediction of treatment adherence [57], prediction of left ventricular filling pressure among HF patients [58] or can reveal relationship between HF symptoms profile and depressiv symptoms [59]. Finally, AI algorithms can predict crucial outcomes in HF management such as mortality and readmission rate.

# Second, interpretability or explainability of the created models becomes a hot topic in the area of machine learning. The end-users are interested not only in the quality of the models but also in understanding the classification process. Some models are easy to understand by their nature (typical examples are decision trees), but some models, typically neural networks, work as a black-box model. To provide some insight into particular decision of the model, several approaches have been proposed. Local Interpretable Model-Agnostic Explanations (LIME) is an interpretability surrogate model which can be used on any black-box model to provide local interpretability for the result of prediction or classification of a single instance [60]. SHapley Additive exPlanations (SHAP) is a game theory-based method to interpreting any machine learning model’s output [61]. It uses the traditional Shapley values from game theory and their related extensions to correlate optimal credit allocation with local explanations. As the expainability issue has not been discussed in the rewieved papers, we can not assess them from this point-of-view.

  1. Can you discuss more the limitations of the current study?  It would be much better to tell what we have not or could not expect from this work.

Thank you for this remark. We have elaborated on that issue in the limitation section.

 Our adjustment:

 Finally, our review focuses mainly on the medical issues of predictive models in HF. We aimed to show  state-of-art achievments in this area, current solutions and level of advancement. We did not consider technical niuances of modelling process such as hyperparameter optimization or tuninng of algorithms. There is increasing need for pragmatic  framework for evaluating clinical utility and validity in ML studies [64]. Development of guidelines for ML implementations in medical research is definitely field for further investigations.

  1. It is really interesting to show what your view about ML or AI in HF could be in the future. 

Thank you for that indication. At the end of discussion section we added paragraph about perspectives of AI techniques

Our adjustment:

The perspective of the AI-based techniques holds great promise for cardiovascular research and clinical practice. Some fields seem especially attractive. Well-trained models can perform real-time, automatized patients’ risk stratification, undertaken at the beginning of the diagnostic path. We can imagine a case when the admitted patient, is classified into one of the pre-established clusters. Affiliation to one of the clusters initially suggests diagnostics, treatment and risk groups. Basing patients’ management on the models, trained on large cohorts, could help physicians, not to overlook some important issues, especially in conditions of overworking and lack of personnel. Further, AI tools can provide more information about the patient, from the same amount of data. Sengupta et al. created a model which can provide an equal amount of information about aortic stenosis severity as if both echocardiography and cardiac CT/MR were performed, by using echocardiography and model only. Such solutions can be widely implemented in less developed areas, where sophisticated diagnostic methods are not available [65]. Another field, which represents great potential is predicting pathology, from seemingly physiological findings, e.g. there are successful attempts to identify patients with a history of atrial fibrillation from the ECG performed during the sinus rhythm[66]. Earlier detection of atrial fibrillation can lead to the vast reduction of vascular changes in the brain and, therefore, improve the patients quality of life.  

  1. More references may need and look at line 58 "e.g.:"  is this correct ?

We appreciate this comment, during the review process we added necessary references and corrected indicated mistake. 

Out adjustment:

  1. Nowak, R.M.; Reed, B.P.; DiSomma, S.; Nanayakkara, P.; Moyer, M.; Millis, S.; Levy, P. Presenting Phenotypes of Acute HeartFailure Patients in the ED: Identification and Implications
  2. Ahmad, T.; Desai, N.; Wilson, F.; Schulte, P.; Dunning, A.; Jacoby, D.; Allen, L.; Fiuzat, M.; Rogers, J.; Felker, G.M.; et al. ClinicalImplications of Cluster Analysis-Based Classification of Acute Decompensated Heart Failure and Correlation with BedsideHemodynamic Profiles.PLoS ONE2016,11, e0145881.
  3. Urban S, Błaziak M, Jura M, Iwanek G, Zdanowicz A, Guzik M, Borkowski A, Gajewski P, Biegus J, Siennicka A, Pondel M, Berka P, Ponikowski P, Zymliński R. Novel Phenotyping for Acute Heart Failure-Unsupervised Machine Learning-Based Approach. Biomedicines. 2022 Jun 27;10(7):1514. doi: 10.3390/biomedicines10071514. PMID: 35884819; PMCID: PMC9313459.
  4. Asyali MH. Discrimination power of long-term heart rate variability measures. Proceedings of the 25th Annual International Conference of the IEEE Engineering in Medicine and Biology Society (IEEE Cat No03CH37439)2003. p. 200-3 Vol.1.

  1. Melillo P, Fusco R, Sansone M, Bracale M, Pecchia L (2011) Discrimination power of long-term heart rate variability measures for chronic heart failure detection. Medical & biological engineering & computing 49:67-74.
  2. Chen W, Liu G, Su S, Jiang Q, Nguyen H. A CHF detection method based on deep learning with RR intervals. 2017 39th Annual International Conference of the IEEE Engineering in Medicine and Biology Society (EMBC)2017. p. 3369-72.
  3. Chen W, Zheng L, Li K, Wang Q, Liu G, Jiang Q (2016) A Novel and Effective Method for Congestive Heart Failure Detection and Quantification Using Dynamic Heart Rate Variability Measurement. PloS one 11:e0165304.
  4. Cikes M, Sanchez-Martinez S, Claggett B, Duchateau N, Piella G, Butakoff C, et al. (2019) Machine learning-based phenogrouping in heart failure to identify responders to cardiac resynchronization therapy. European journal of heart failure 21:74-85.
  5. Graven LJ, Higgins MK, Reilly CM, Dunbar SB (2018) Heart Failure Symptoms Profile Associated With Depressive Symptoms. Clinical nursing research:1054773818757312.
  6. Ribeiro, M.T., Singh, S., Guestrin, C.: " why should i trust you?" explaining the predictions of any classifier. In: Proceedings of the 22nd ACM SIGKDD International Conference on Knowledge Discovery and Data Mining, pp. 1135{1144 (2016)
  7. Lundberg, S.M., Erion, G., Chen, H., DeGrave, A., Prutkin, J.M., Nair, B., Katz, R., Himmelfarb, J., Bansal, N., Lee, S.-I.: From local explanations to global understanding with explainable ai for trees. Nature Machine Intelligence 2(1), 2522{5839 (2020)                    62. Duchnowski P, Hryniewiecki T, Koźma M, Mariusz K, Piotr S. High-sensitivity troponin T is a prognostic marker of hemodynamic instability in patients undergoing valve surgery. Biomark Med. 2018 Dec;12(12):1303-1309. doi: 10.2217/bmm-2018-0186. Epub 2018 Dec 6. PMID: 30520660.     63. Duchnowski P, Hryniewiecki T, Kuśmierczyk M, Szymański P. Postoperative high-sensitivity troponin T as a predictor of sudden cardiac arrest in patients undergoing cardiac surgery. Cardiol J. 2019;26(6):777-781. doi: 10.5603/CJ.a2019.0005. Epub 2019 Jan 31. PMID: 30701514; PMCID: PMC8083033.                                                                                                                                64. Ghazi L, Ahmad T, Wilson FP. A Clinical Framework for Evaluating Machine Learning Studies. JACC Heart Fail. 2022 Aug 5:S2213-1779(22)00407-3. doi: 10.1016/j.jchf.2022.07.002. Epub ahead 11

Reviewer 2 Report

I am really grateful for reviewing this manuscript. In my opinion, this manuscript can be published once some revision is done successfully. This study reviewed 9 articles on heart failure to analyze their machine learning approaches and areas under the receiver operating characteristic curves (AUCs) in training and validation sets, which came from different sources. I would like to point out that this is a great achievement. However, it needs to be noted that emerging literature requests due attention to the importance of data interpretation and explainable AI, i.e., variable importance and SHapley Additive exPlanations (SHAP). I would like to suggest the authors to address this issue throughout the manuscript. 

Author Response

We would like to thank the Editors and the Reviewers for their valuable comments. We have amended our manuscript accordingly. The detailed responses to all issues raised by the Reviewers are provided below.

Reviewers' comments:

I am really grateful for reviewing this manuscript. In my opinion, this manuscript can be published once some revision is done successfully. This study reviewed 9 articles on heart failure to analyze their machine learning approaches and areas under the receiver operating characteristic curves (AUCs) in training and validation sets, which came from different sources. I would like to point out that this is a great achievement. However, it needs to be noted that emerging literature requests due attention to the importance of data interpretation and explainable AI, i.e., variable importance and SHapley Additive exPlanations (SHAP). I would like to suggest the authors to address this issue throughout the manuscript. 

Thank you for that remark. We have added a paragraph in discussion about existing approaches describing interpretability or explainability of the AI models.

Our adjustment:

Second, interpretability or explainability of the created models becomes a hot topic in the area of machine learning. The end-users are interested not only in the quality of the models but also in understanding the classification process. Some models are easy to understand by their nature (typical examples are decision trees), but some models, typically neural networks, work as a black-box model. To provide some insight into particular decision of the model, several approaches have been proposed. Local Interpretable Model-Agnostic Explanations (LIME) is an interpretability surrogate model which can be used on any black-box model to provide local interpretability for the result of prediction or classification of a single instance [60]. SHapley Additive exPlanations (SHAP) is a game theory-based method to interpreting any machine learning model’s output [61]. It uses the traditional Shapley values from game theory and their related extensions to correlate optimal credit allocation with local explanations. As the expainability issue has not been discussed in the rewieved papers, we can not assess them from this point-of-view.

Reviewer 3 Report

Thank you for the opportunity to review your manuscript entitled "Artificial intelligence approach in guiding management of heart failure patients - predictive models. A systematic review".

Troponin T (TnT) is a polypeptide that is part of the striated contractile muscle apparatus. A very important aspect from a diagnostic point of view is the fact that the sequence of troponins of cardiac origin differs from the sequence of skeletal troponins. Thanks to this, after obtaining specific monoclonal antibodies, it became possible to use them in the diagnosis of ischemia and hypoxia of cardiomyocytes [1,2]. In the available literature, numerous articles describe high sensitivity troponin T (hs-TnT) as a biomarker of predictive importance in various diseases of the cardiovascular system, such as heart failure or sudden cardiac arrest [1,2].

Abstract, title and references.

The aim of the study is clear. The title is informative and relevant.

The references are relevant, recent, and referenced correctly.

Introduction.

It is clear. The research question is clearly outlined.

Results and Discussion.

The results are discussed from multiple angles and placed into context without being overinterpreted. The conclusions answer the aims of the study. The conclusions supported by references and results. The limitations of the study are opportunities to inform future research.

Overall. The study design was appropriate to answer the aim. The manuscript is well written and a stimulus for the readership.

Minor revisions:

Does the model take into account the parameter of heart damage, which is Troponin T?

Please add the following reference: 

1. DOI: 10.2217/bmm-2018-0186

2. DOI: 10.5603/CJ.a2019.0005

Author Response

Thank you for this remark. We have elaborated on that issue in the paragraph about the used predictive features in discussion section.

 Our adjustment:

Third, described models used only variables, which were collected in datasets used as derivation cohorts. Three of all included studies used well-established prognostic marker such as NT-proBNP as a predictive feature, but only one model used troponin to predict outcome [62-63]. There exists other, not included parameters, which can be also important prognostic factors of cardiovascular diseases and can potentially improve the model performance [40-41].

Round 2

Reviewer 1 Report

Thank you for the revision.  They have addressed mostly my concerns.